# Co-Sleeping as a Protector against Malocclusion in the Primary Dentition: A Cross-Sectional Study

**DOI:** 10.3390/jcm11092338

**Published:** 2022-04-22

**Authors:** María Carrillo-Díaz, Ana Ruiz-Guillén, María Moya, Martín Romero-Maroto, María José González-Olmo

**Affiliations:** 1Department of Paediatric Dentistry, Rey Juan Carlos University, 28922 Alcorcón, Spain; ana.guillen@urjc.es (A.R.-G.); maria.moya@urjc.es (M.M.); 2Department of Orthodontics, Rey Juan Carlos University, 28922 Alcorcón, Spain; martin.romero@urjc.es (M.R.-M.); mariajose.gonzalez@urjc.es (M.J.G.-O.)

**Keywords:** breastfeeding, co-sleeping, malocclusion, oral health, child

## Abstract

Mothers practice co-sleeping and breastfeeding simultaneously, for convenience and to strengthen bonding. Due to the scarcity of studies analyzing the impact of co-sleeping on primary dentition, this study aimed to analyze the possible effects of co-sleeping on children’s occlusion. In this cross-sectional study, mothers of 221 children aged 2–5 years who had been breastfed for less than 6 months completed a questionnaire about non-nutritive sucking habits. The WHO (World Health Organization) and IOTN-AC indices (the Aesthetic Component of the Index of Orthodontic Treatment Need) were used to assess malocclusion. The type of sagittal (dental and skeletal), transverse and vertical malocclusion was recorded. The non-co-sleeping group showed significantly higher pacifier use (*p* < 0.05), digital sucking (*p* < 0.05) and atypical swallowing (*p* < 0.05) habits. The non-co-sleeping group showed significantly higher mean scores on the IOTN-AC (*p* < 0.05) and WHO (*p* < 0.01), a significantly higher presence of canine class II (*p* < 0.05), anterior open bite (*p* < 0.05), posterior crossbite (*p* < 0.05), overbite (*p* < 0.05), skeletal class II (*p* < 0.01) and protrusion (*p* < 0.05). In conclusion, children who practice co-sleeping appear to have a lower frequency and duration of non-nutritive sucking habits. Co-sleeping may contribute to a lower development of malocclusions in children who are weaned early (before six months of age).

## 1. Introduction

The American Academy of Pediatric Dentistry (2019) notes that the primary dentition begins in infancy with the eruption of the first tooth, usually at six months of age, and is completed at approximately three years of age; this forms the basis for the development of the permanent dentition by determining the spacing and occlusion of future developing teeth [1]. For this reason, malocclusion in the primary dentition is considered an important risk factor for the appearance of occlusal disorders in both the mixed dentition and the permanent dentition [2,3]. Malocclusion is defined as a developmental disorder of the maxillofacial system resulting from genetic and environmental factors [4], which can cause masticatory, digestive, phonation, swallowing [5] and periodontal problems, caries [6], headaches, muscle pain, esthetic problems, low self-esteem, bullying [7], etc. In short, it can have a negative impact on the quality of life of the individual [8]. In addition, recent studies show that it affects 1 out of every 2 children/adolescents [9]; malocclusion in early childhood is higher, affecting 53.4% of this population [10].

Therefore, malocclusion is considered a public health problem of great interest for research, due to the serious consequences it entails and its high prevalence. Therefore, considering that its etiology can be modified, knowledge of the environmental factors involved in its appearance can help prevent or manage this condition more successfully.

The environmental factors involved in the etiology of malocclusions are classified into two groups: nutritive sucking habits (breastfeeding and bottle feeding) and non-nutritive sucking habits (pacifier and digital sucking), whose main function is to provide the need for sucking, calm and security to the baby [11]. Non-nutritive sucking habits have been described as precursor risk factors for an anterior open bite and posterior crossbite [12,13]. However, breastfeeding, apart from the innumerable systemic, psychological, immunological and nutritional benefits [13,14,15,16,17,18], also stands out for its benefits at the oral level [19,20]. The sucking movements that occur during breastfeeding involve peristaltic movements of the tongue around the nipple, which helps guide the morphology of the palate by rounding and flattening it; there is also a development of the perioral musculature, necessary for efficient swallowing and phonation [21,22,23,24]. Exclusive breastfeeding is associated with a lower probability of developing a class II incisal relationship. Children who are exclusively breastfed for more than 6 months have also been found to have wider intermolar spaces and a lower occurrence of harmful oral habits [19]. The action of breastfeeding uses intense muscular activity and benefits oral motor development.

On the other hand, the performance of co-sleeping, which refers to a mother and child sharing a bed, sharing a room, or being physically close [25] during the infant’s sleep phase, is a common practice in different cultures [25,26] and provides the infant with better thermoregulation and good cardiac and respiratory balance [27]. In addition, there is evidence that for nighttime feeding, a high number of mothers simultaneously practice co-sleeping and breastfeeding, for comfort and bonding reinforcement.

Numerous authors agree that too short a duration of breastfeeding (less than 6 months), or no breastfeeding at all, could lead to the use of more non-nutritive sucking habits, altering facial balance and giving rise to malocclusions, such as an anterior open bite and posterior crossbite in the primary dentition [13,16,17,18,21,24]. That is why, in this context, it is important to promote breastfeeding [14], since the psychological and affective needs of the child are more satisfied; therefore, the child is calmer and it is less necessary to use the pacifier, digital suction or other objects [17,21]. However, sometimes breastfeeding ceases before 6 months of life. In these children, co-sleeping could favor the security of the child, due to the action of sleeping next to the mother, contributing to the maintenance of calm [28] and avoiding the appearance of non-nutritive sucking habits, thus protecting the child against malocclusions in the primary dentition.

Therefore, the aim of this study was to shed light on existing gaps in the literature in relation to the possible effects of the practice of co-sleeping on the oral health of a child. Specifically, the objectives of this study were:-To assess whether the practice of co-sleeping is influenced by sociodemographic variables;-To analyze whether the practice of co-sleeping has an impact on non-nutritive sucking habits, their duration and frequency;-To study whether the coincidence between the onset of these habits and the time of weaning is greater in subjects who do not co-sleep;-To investigate whether co-sleeping alters the relationship between non-nutritive sucking habits and malocclusion in children who are weaned early.

## 2. Materials and Methods

### 2.1. Study Design and Setting

A cross-sectional study was carried out from September 2019 to March 2020. The research team at the Department of Pediatric Dentistry and Orthodontics of the Universidad Rey Juan Carlos contacted 9 public nursery schools in the community of Madrid (Spain), inviting them to participate in the study and, in exchange, offering an oral report for the child and a talk on oral health promotion and disease prevention. The kindergartens were randomly selected from the southern part of the region. Through the schools that agreed to participate, the mothers were sent to an appointment for the dental check-up.

A total of 221 children were examined, belonging to varied sociodemographic strata, with representation from all socioeconomic levels. This was a convenience sample but with an optimal sample size that was based on the results of the prevalence of malocclusion in the primary dentition of 54% [9]. A random sample of 221 individuals was estimated to be a representative number for the 3-year cohort, with a confidence level of 95% based on a degree of precision of ±4%. The value of about 221 individuals would be optimal if this were a randomized study, but since this was a convenience sample, this value was taken as a reference. 

The inclusion criteria were as follows: children aged 2 to 5 years who had been breastfed for a period of less than 6 months. To guarantee a healthy sample selection, preterm infants were excluded, as were those who required instruments (forceps, etc.) during delivery, children with any local/systemic condition that could affect their oral health status and children with teeth with anomalies in number, size and/or shape. Mothers who refused to participate in the study, were not fluent in Spanish, had mental or physical disabilities preventing them from answering the questions or submitted an incomplete questionnaire were excluded. The sample was divided into two groups: mother–child dyads that practiced co-sleeping daily and those that did not share a bed (except at specific times). No other inclusion/exclusion criteria were used.

All mothers who agreed to participate in the study signed an informed consent form. Ethical approval for this study was obtained from the Ethics Committee for Research Affairs of the Universidad Rey Juan Carlos (2409201913019).

### 2.2. Measures

Data were collected by means of a questionnaire that the mothers filled out upon arrival at the clinic. Instructions for filling it out were provided by a member of the research team. Data were recorded on gender and age of the child (months), mother’s age, maternal education (no education/primary, secondary/higher), maternal occupation (employed/unemployed) and family socioeconomic status (low, low-middle, medium, medium-high, medium-high and high). Bottle use, pacifier use, digital sucking and oral respiration (yes/no) were recorded. If yes, the frequency of the habit was recorded in hours per day and the duration of the habit in months.

The data were collected from regular individual visits of pediatric patients attending a clinic that offers regular dental treatments such as check-ups, caries treatment, and orthodontics for children and adults. Participants in this study received a free oral health check-up. The oral examination was performed by a single orthodontist in a seated position using basic oral examination tools (mirror, flexible millimeter ruler, latex gloves, masks and goggles); a dental hygienist simultaneously filled out the clinical examination form. The index recommended by the World Health Organization (WHO) was used to evaluate the presence and severity of malocclusion. This index considers certain occlusal parameters of the teeth and the relationship between the maxillary and mandibular arches [29,30]. This index has three categories: no malocclusion (absence of any alteration), mild pathological occlusion (one or more teeth with rotation or slight crowding or spacing) and occlusion with moderate/severe pathology (unesthetic effect on facial esthetics, significant reduction in mastication or phonetic problems caused by one or more of the following conditions affecting all 4 incisors: overjet > 9 mm, anterior crossbite equal to or greater than the size of 1 tooth, open bite, midline deviation estimated to be 4 mm or more and crowding or spacing estimated to be 4 mm or more) [29,30]. In turn, malocclusion was evaluated specifically at the sagittal (class I, II or III canine; terminal plane, mesial or distal step in the primary dentition and overjet), transverse (crossbite and normoocclusion) and vertical (overbite, anterior open bite and normoocclusion) levels. In addition, the patient’s right profile was selected in a natural postural head position, with gaze towards the horizon, lips at rest and teeth in maximum intercuspidation. A true vertical was traced through the subnasal point to record the skeletal class (I, II or III).

Finally, the same orthodontist evaluated the degree of malocclusion with the Orthodontic Treatment Need Index (OTNI). The esthetic component of the IOTN (IOTN-AC) is evaluated through a series of 10 intraoral frontal color photographs corresponding to 10 possible levels of dental esthetics. The scores range from 1 (best esthetic appearance) to 10 (worst). The IOTN-AC is an index used to determine the patient’s need for orthodontic treatment in terms of dental esthetics [31]. This index has previously been used in children of a similar age.

A pilot survey was conducted with 5 mothers who subsequently participated in the study. A researcher asked them how clear the instructions were, or which questions were difficult to understand or answer. Some of the questions were rephrased for better understanding by the participants to avoid possible misunderstanding biases.

### 2.3. Statistical Analysis

The statistical program SPSS (Statistical Package of the Social Sciences Program for Windows 27.0) was used for statistical calculations. A χ^2^ test was used to assess the difference in maternal age, maternal employment, educational and socioeconomic status as well as the presence of non-nutritive sucking habits in the co-sleeping and non-co-sleeping groups. A Student’s *t*-test was used to investigate whether co-sleepers had a shorter duration and frequency of non-nutritive sucking habits. Cohen’s d statistic was used to assess the effect size. Pearson’s correlation coefficient was used to analyze the association between continuous variables. In addition, a two-way ANOVA was performed to assess the interaction of the variables of co-sleeping and pacifier use on the IOTN-AC. The results are expressed as means ± standard deviation, and percentages and differences were considered significant at the *p* level < 0.05.

## 3. Results

### 3.1. Sociodemographic Analysis

Two hundred and twenty-one mother–child pairs were involved in the study: 118 (53.4%) males and 103 (46.6%) females. The mean age (±SD) of the mothers was 32.31 (±4.86) years, while that of the children was 3.9 (±0.87) years. (Table 1). Co-sleeping was practiced by 130 subjects (58.8%) and not practiced by 91 subjects (41.2%). The mean time and SD of breastfeeding in months was 4.42 (±0.72). No differences were found in terms of co-sleeping (yes or no), according to maternal age (≤30 or >31 years old) (χ^2^ (1) = 0, *p* = 1, educational level (χ^2^ (1) = 2.29, *p* = 0.13), maternal job (χ^2^ (1) = 0.59, *p* = 0.44) or socioeconomic level (χ^2^ (4) = 3.41, *p* = 0.49). See Table 1.

### 3.2. Co-Sleeping and Habits

Of the children, 110 children used pacifiers (49.7%), 47 had a digital sucking habit (21.2%) and 30 subjects (13.5%) had atypical swallowing. The group that did not co-sleep showed a significantly higher habit of using pacifiers (χ^2^ (1) = 4.43, *p* = 0.03), digital sucking χ^2^ (1) = 6.52, *p* = 0.01 and atypical swallowing χ^2^(1) = 5.07, *p* = 0.02. No significant differences were found for oral respiration (χ^2^ (1) = 0.44, *p* = 0.83).

As can be seen in Table 2, the mean duration of the habits was significantly longer for those who did not co-sleep, for pacifier habit (*p* < 0.05) and for digital sucking (*p* < 0.01). The frequency of habit use throughout the day was also significantly lower in the co-sleeping group (*p* < 0.05).

Regarding the onset of pacifier and digital sucking habits, differences were found in the two groups. The onset of the pacifier habit coincided with the time of weaning in the co-sleeping group in 10.5% of the subjects, while for those who did not co-sleep the percentage was 26.41%, this coincidence not being significant: χ^2^ (2) = 4.92, *p* = 0.08. Regarding the digital sucking habit, something similar happens: this coincidence occurs in co-sleeping and non-co-sleeping groups at a rate of 10 and 22%, respectively. This coincidence is not significant: χ^2^ (2) = 1.21, *p* = 0.54.

As can be seen in Table 3, daily pacifier frequency (*p* < 0.01) and pacifier use duration (*p* < 0.01) correlate with the IOTN-AC index; likewise, digital sucking frequency (*p* < 0.01) and digital sucking duration (*p* < 0.01) correlate with the WHO index.

### 3.3. Breastfeeding and Malocclusion

The non-co-sleeping group showed significantly higher mean scores in the IOTN-AC (*p* < 0.05) and WHO (*p* < 0.01) indices (see Table 4). Regarding specific orthodontic malocclusions, there was a significantly higher presence of canine class II (χ^2^ (2) = 8.57, *p* = 0.01; *p* < 0.05), anterior open bite (χ^2^ (1) = 4.37, *p* = 0.03; *p* < 0. 05), posterior crossbite (χ^2^ (1) = 3.95, *p* = 0.04; *p* < 0.05), overbite (χ^2^ (1) = 6.42, *p* = 0.01; *p* < 0.05), skeletal class II (χ^2^ (1) = 9.81, *p* = 0.007; *p* < 0.01) and protrusion (χ^2^ (1) = 7.26, *p* = 0.026; *p* < 0.05). No significant difference was found for mesial step, distal step or flush terminal plane.

In the interaction analysis of the variables of co-sleeping and pacifier use on the IOTN-AC, a significant interactive relationship was found (F(1,217) = 5.98; *p* = 0.015; η^2^ = 0.027) (see Figure 1).

The same occurred for digital sucking, co-sleeping and the IOTN-AC (F(1,217) = 3.96; *p* = 0.048; η^2^ = 0.018) (see Figure 2).

## 4. Discussion

The results of this study show that children who breastfeed less than 6 months but co-sleep have less risk of using pacifier, digital sucking and oral breathing habits as opposed to when early weaning occurs and co-sleeping is not practiced. To the best of our knowledge, there are no studies that have analyzed these associations, but the results obtained ratify the hypotheses raised, since co-sleeping could reduce the need to incorporate the pacifier in early weaning (before 6 months of age) by providing a sense of security, physiological calm and comfort to infants, similar to that facilitated by the pacifier or digital suction, as reported in previous studies [28].

The results of the present study were generally consistent with earlier studies that claim that the prevalence of non-nutritive sucking habits is higher in infants who breastfeed for less than 6 months [16,17,18,24]. Agarwal et al. (2014) [24] supported these findings by reporting that if breastfeeding occurs for less than 6 months, it is doubly likely that a non-nutritive sucking habit will develop, which will alter the intra-arch transverse dimensions, resulting in posterior crossbites, lack of space for the teeth in the maxilla and anterior open bites [32]. Other studies that analyzed non-nutritive sucking habits taking into account the breastfeeding variable also concluded that the durations of pacifier and digit habits were each positively related to the prevalence of certain malocclusions, agreeing that prolonged breastfeeding decreases the risk factors for the appearance of malocclusion [33]. It has also been shown in previous studies that there is a higher prevalence of malocclusion in children who were never breastfed and who always used a pacifier to sleep [10]; however, the results of the present study indicate that children who breastfeed less than 6 months but co-sleep are at a lower risk of malocclusion (WHO and IOTN-AC). This is a consequence of minimizing the use of non-nutritive sucking habits. Previous work has shown that pacifier use is directly associated with digital sucking, non-nutritive habits detrimental to occlusion and the development of malocclusions in the primary dentition [33,34].

Some previous research has shown that non-nutritive sucking habits and their duration are more important in the development of malocclusions than the type of breastfeeding [14,33], that non-nutritive sucking habits are related to class II malocclusion [15] and that children who use pacifiers or perform digital sucking are 4 times more susceptible to have overbites [35]. On the other hand, the absence of these habits reduces the prevalence of posterior crossbites by almost 50% [16]. In the same line of results, Paolantonio et al. (2019) [36], in their study carried out in children of similar ages to the present study, observed that children with non-nutritive sucking habits showed a high prevalence of class II malocclusion, anterior open bite and posterior crossbite due to pacifier use and digital sucking, coinciding with the results found in the present study.

It is also important to recognize some limitations of this study. First, we analyzed the role played by a set of variables such as the practice of co-sleeping, breastfeeding and non-nutritive sucking habits, but these represent only some of the many factors that could act in the development of malocclusions. For example, genetics and chronic diseases that may be associated with malocclusions were not considered. Second, we used a convenience sample, which came from a specific segment of the population of children in the community of Madrid, and this could limit the possibilities of generalizing the results. A possible third limitation comes from the use of self-report measures, which can be affected by recall bias and responses based on social desirability. Fourth, due to the cross-sectional design of the study, it is possible that mothers’ recall of their breastfeeding experiences, co-sleeping and non-nutritive sucking habits may be incomplete or inaccurate and, furthermore, do not allow causal relationships to be made. Fifth, the diagnosis of malocclusions was made clinically without the performance of complementary tests such as radiographs or study models, which could have biased some diagnoses. Finally, a single observer performed the dental examination, so verification of intra-examiner reliability was not performed.

This study has important practical applications for oral infant care. All health professionals, including dentists, have a responsibility to protect and promote breastfeeding by supporting WHO recommendations and to provide correct and up-to-date messages based on scientific evidence. From this study we can extract some guidelines in terms of oral health promotion: if the mother believes that she will not be able to breastfeed for a minimum period of 6 months, the recommendation is that she should practice safe co-sleeping to avoid the appearance of non-nutritive sucking habits and thus minimize the risk of malocclusions. Conversely, if a mother plans to prolong exclusive breastfeeding for more than 6 months, findings in previous studies note that pacifier use considerably decreased without being associated with the practice of co-sleeping [32].

Future lines of research are required to utilize longitudinal studies to assess whether malocclusions originating from such factors in the primary dentition are likely to lead to disorders in the mixed and definitive dentitions.

## 5. Conclusions

From the results extracted, it can be concluded that in children who are weaned early (before six months of age):-Co-sleeping is not influenced by sociodemographic variables.-Children who practice co-sleeping appear to have a lower frequency and duration of non-nutritive sucking habits.-In addition, co-sleeping may contribute to a reduced development of malocclusions.-Co-sleeping appears to act as a moderator in the relationship between non-nutritive sucking habits and malocclusion.

## Figures and Tables

**Figure 1 jcm-11-02338-f001:**
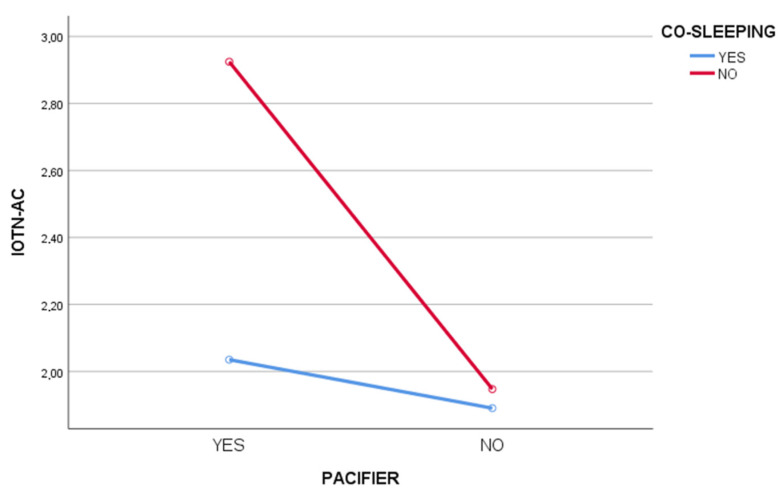
Two-way ANOVA for the variables of co-sleeping (yes/no) and pacifier (yes/no) on IOTN-AC.

**Figure 2 jcm-11-02338-f002:**
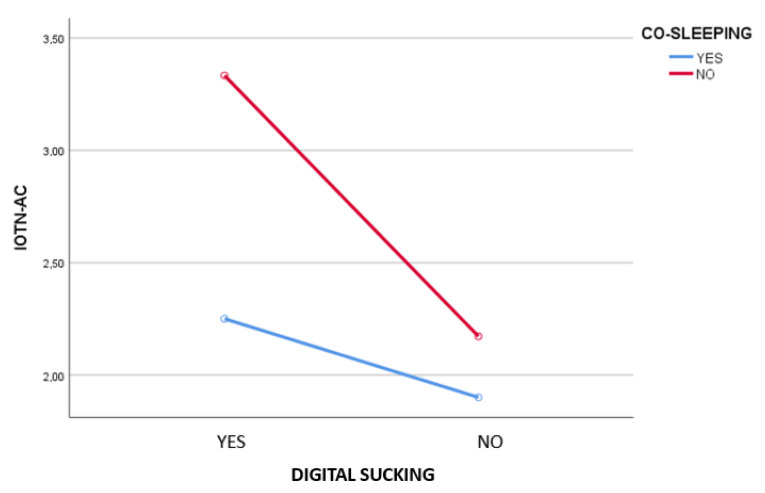
Two-way ANOVA for the variables of co-sleeping (yes/no) and digital sucking (yes/no) on IOTN-AC.

**Table 1 jcm-11-02338-t001:** Demographic characteristics of participants.

Variables	Co-Sleeping	Non-Co-Sleeping	*n* (%)
**Gender**			
Male	69	49	118 (53.4%)
Female	61	42	103 (46.6%)
Total *n* (%)	130	91	221 (100%)
**Children’s age**			
2 years	4	5	9 (4.1%)
3 years	45	25	70 (31.7%)
4 years	42	34	76 (34.4%)
5 years	39	27	66 (29.9%)
Total *n* (%)	130	91	221 (100%)
**Mother’s age range**			
≤30	40	28	68 (30.8%)
>31	90	63	153 (69.2%)
Total *n* (%)	130	91	221 (100%)
**Mother’s educational level**			
No studies or primary	27	27	54 (24.4%)
Secondary and higher education	103	64	167 (75.6%)
Total *n* (%)	130	91	221 (100%)
**Mother’s work**			
Employee	85	64	149 (67.4%)
Unemployed	45	27	72 (32.6%)
Total *n* (%)	130	91	221 (100%)
**Socioeconomic level**			
Low	16	17	33 (14.9%)
Low-medium	34	17	51 (23.1%)
Medium	40	30	70 (31.7%)
Medium-high	30	18	48 (21.7%)
High	10	9	19 (8.6%)
Total *n* (%)	130	91	221 (100%)

**Table 2 jcm-11-02338-t002:** Comparison of daily frequency and duration of oral habits (pacifier and digital sucking) in co-sleeping (*n* = 130) and non-co-sleeping (*n* = 91) groups.

Variables	Co-Sleeping M (SD)	Non-Co-Sleeping M (SD)	*p*-Value	D Cohen
**Pacifier sucking**				
Daily frequency (hours in a day)	5.94 (5.69)	8.77 (6.40)	0.016 *	0.46
Duration (months)	18.91 (10.88)	24.86 (9.82)	0.043 *	0.57
**Digital sucking**				
Daily frequency (hours in a day)	2.90 (1.44)	3.92 (1.61)	0.029 *	0.66
Duration (months)	17.25 (12.75)	25.11 (12.79)	0.003 **	0.61

Note: Student’s *t*-test, comparing results between the co-sleeping and non-co-sleeping group. * = *p* level < 0.05. ** = *p* < 0.01.

**Table 3 jcm-11-02338-t003:** Pearson’s correlation between pacifier (duration and daily frequency), digital sucking (duration and daily frequency), IOTN-AC and WHO index.

		1	2	3	4	5	6
**Daily frequency of pacifier use (hours)**	r		0.210	0.147	0.000	0.194	0.230
	*p*		0.006	0.246	0.998	0.012	0.003
	*n*		168	64	64	168	168
**Pacifier duration (months)**	r			−0.041	0.129	0.210	0.290
	*p*			0.750	0.311	0.006	0.000
	*n*			64	64	168	168
**Daily frequency of digital sucking (hours)**	r				0.362	0.157	0.045
	*p*				0.001	0.147	0.682
	*n*				87	87	87
**Duration of digital sucking (months)**	r					0.146	0.061
	*p*					0.176	0.572
	*n*					87	87
**IOTN-AC**	r						0.573
	*p*						0.000
	*n*						221
**WHO INDEX**	r						
	*p*						
	*n*						

**Table 4 jcm-11-02338-t004:** Comparison of clinical variables of malocclusion (IOTN-AC and WHO) in co-sleeping (*n* = 130) and non-co-sleeping (*n* = 91) groups.

Variables	Co-Sleeping M (SD)	Non-Co-Sleeping M (SD)	*p*-Value	D Cohen
WHO	1.81 (0.71)	2.27 (0.76)	0.000 *	0.62
IOTN-AC	1.95 (1.04)	2.38 (1.45)	0.011 **	0.34

Note: Student’s *t*-test, comparing results between the co-sleeping and non-co-sleeping group. * = *p* level < 0.05. ** = *p* level < 0.01.

## Data Availability

The data that support the findings of this study are available on request from the corresponding author. The data are not publicly available due to privacy and ethical restrictions.

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
