# Peer review of "Co-Sleeping as a Protector against Malocclusion in the Primary Dentition: A Cross-Sectional Study"

_jcm, 2022, doi:10.3390/jcm11092338_

Round 1

Reviewer 1 Report

I would like to congratulate authors for this interesting paper. 

Manuscript has been prepared properly, according to science's papers requirements.

Well-described methodology of the research is followed by the adequate discussion over the results of the analyses.

 But some improvements are suggested. I provided  more details below.

  • Abstract of the manuscript should be supplemented with the brief information concerning the novelty of the studies. 
  • The results are adequately described. If you have same data on figures and in tables than omit one of them.
  • Discussion - this section  should be supplemented with brief information on the actually known habbits and reasons that provides to malocclusions and compare it with obtained results.

Author Response

Reviewer #1: Review for 1667251

Full Title: CO-SLEEPING AS A PROTECTOR AGAINST MALOCCLUSION IN THE PRIMARY DENTITION: A CROSS-SECTIONAL STUDY.

Abstract of the manuscript should be supplemented with the brief information concerning the novelty of the studies.

We proceeded to specify the absence of previous studies on the impact of co-sleeping on malocclusion, thus justifying the novelty of our results.

The results are adequately described. If you have same data on figures and in tables than omit one of them.

We completely agree. The information already detailed in the Table 1 has been removed from the text. In addition, Table 2 has been eliminated, since it contained information explained in the text.

Discussion - this section should be supplemented with brief information on the actually known habits and reasons that provides to malocclusions and compare it with obtained results.

Following the reviewer's suggestions, a search was made for articles analyzing the habits that produce malocclusion in children and the information was contrasted with the results of our study.

Reviewer 2 Report

Dear authors,

I suggest that you refer to your recent publication:

Carrillo-Díaz M, Ortega-Martínez AR, Ruiz-Guillén A, Romero-Maroto M, González-Olmo MJ. The impact of co-sleeping less than 6 months on children's anxiety, oral habits, and malocclusion in a Spanish sample between 2 and 5 years old: a cross-sectional study. Eur J Orthod. 2022;44(1):110-115.

The current study seems very much similar or perhaps in its greatest part a repetition of the published one. There are also many problems in the submitted manuscript:

1-You report that you have included 2-4 year-old children while Table 1 shows that 5-year-old children were also included.

2-Materials and Methods: the basis for the selection of the nurseries is not mentioned.

3- All questions included in the questionnaire including the sociodemographic questions should be described.

4-Measures: there is no mention of the Intra-examiner reliability.

5-Statistical analysis: this section, in particular, should be checked by a biostatistician. According to your study, co-sleeping was categorized as being practiced or not, use of t-test to assess the association with sociodemographic profile is inappropriate. For t-test, you need a continuous dependent variable. Also, to use two- way ANOVA you need a continuous dependent variable.  The IOTN-AC is NOT a continuous variable, so you should consult a biostatistician on the alternative test for non-parametric data, if there is one.

6- Results: the reported percentages for boys and girls in the text and in Table 1 should be checked.

7-Results: how the different socioeconomic levels were categorized should be described.

8-Results of co-sleeping and habits+ Figures 1 and 2 and the relevant text should be checked by a biostatistician.

9-Results line 103: “There were 110 pacifiers…”: what is that supposed to mean?

10-Line 256: “straight step”: there is no such term.

11-Conclusions: the conclusion should be shaped according to the results of the study. Also, the authors should keep in mind the aims of their study when they write the conclusion.

Author Response

I suggest that you refer to your recent publication:

Carrillo-Díaz M, Ortega-Martínez AR, Ruiz-Guillén A, Romero-Maroto M, González-Olmo MJ. The impact of co-sleeping less than 6 months on children's anxiety, oral habits, and malocclusion in a Spanish sample between 2 and 5 years old: a cross-sectional study. Eur J Orthod. 2022;44(1):110-115.

This research group has been studying the impact of co-sleeping on oral health (caries, malocclusions and quality of life) for several years. Although they share the same subject matter, the sample, variables and objectives of each study are different. Specifically, the objective of the study you mentioned was:

The purpose of this study was to investigate whether the early termination of co-sleeping is associated with high levels of anxiety, non-nutritive sucking habits for self-comfort, and increased malocclusion.

The current study seems very much similar or perhaps in its greatest part a repetition of the published one. There are also many problems in the submitted manuscript:

1-You report that you have included 2-4 year-old children while Table 1 shows that 5-year-old children were also included.

The material and methods section has been changed from 2-4 year-old to 2-5 year-old.

2-Materials and Methods: the basis for the selection of the nurseries is not mentioned.

A random sampling of nursery schools in the southern part of the region was carried out. In addition, in line 98 it is mentioned that it is a convenience sample, knowing that this represents a limitation for the study, also described in the limitations.

3- All questions included in the questionnaire including the sociodemographic questions should be described.

As suggested by the reviewer, it has been described in the material and method section.

4-Measures: there is no mention of the Intra-examiner reliability.

Line 144 specifies that there is only one examiner. “Finally, the same orthodontist evaluated the degree of malocclusion with the Orthodontic Treatment Need Index (IOTN).” We have considered that this could be a limitation and have therefore included it in the limitations section of the study.

5-Statistical analysis: this section, in particular, should be checked by a biostatistician. According to your study, co-sleeping was categorized as being practiced or not, use of t-test to assess the association with sociodemographic profile is inappropriate. For t-test, you need a continuous dependent variable.

A χ2 test was performed to assess the difference in maternal age, maternal employment, educational and socioeconomic status in the co-sleeping and non-co-sleeping groups. This has been added to the results section.

Also, to use two- way ANOVA you need a continuous dependent variable.  The IOTN-AC is NOT a continuous variable, so you should consult a biostatistician on the alternative test for non-parametric data, if there is one.

Dear reviewer, we appreciate your great analysis of the results. The variable IOTN-AC is a variable that could be discrete, as it is a variable that has a countable number of values. However, the discrete variable has many levels (10), so we have decided to treat it as a continuous variable. The IOTN-AC variable has already been used in previous literature as a continuous variable.

Bellot-Arcís C, Montiel-Company JM, Almerich-Silla JM. Psychosocial impact of malocclusion in Spanish adolescents. Korean J Orthod. 2013 Aug;43(4):193-200. doi: 10.4041/kjod.2013.43.4.193. Epub 2013 Aug 22. PMID: 24015389; PMCID: PMC3762961.

Bellot-Arcís C, Montiel-Company JM, Pinho T, Almerich-Silla JM. Relationship between perception of malocclusion and the psychological impact of dental aesthetics in university students. J Clin Exp Dent. 2015 Feb 1;7(1):e18-22. doi: 10.4317/jced.52157. PMID: 25810834; PMCID: PMC4368010.

6- Results: the reported percentages for boys and girls in the text and in Table 1 should be checked.

The gender percentages have been modified in the table and in the text.

7-Results: how the different socioeconomic levels were categorized should be described.

As suggested by the reviewer, it has been described in the material and method section and in table 1.

8-Results of co-sleeping and habits+ Figures 1 and 2 and the relevant text should be checked by a biostatistician.

The answer to this suggestion can be found in the point 5.

9-Results line 103: “There were 110 pacifiers…”: what is that supposed to mean?

It is paragraph 203. It is a translation error that has been solved.

10-Line 256: “straight step”: there is no such term.

"Straight step" has been replaced by "flush terminal plane" in the manuscript.

11-Conclusions: the conclusion should be shaped according to the results of the study. Also, the authors should keep in mind the aims of their study when they write the conclusion.

The conclusions have been modified according to the reviewer's suggestions.

Round 2

Reviewer 2 Report

Dear authors,

Thank you for the effort spent to improve the manuscript. The manuscript has improved; however; it is important that you also address the following points:

1-The abstract should be modified according to the changed manuscript (e.g. age group of children and conclusions).

2-Introduction lines 52-54: “However, breastfeeding, apart from the innumerable systemic, psychological, immunological and nutritional benefits [13-18], also stands out for its benefits at the oral level [19].”

Support these sentences with the below recent reference:

Risk Factors of Early Childhood Caries Among Preschool Children in Eastern Saudi Arabia. Science Progress 2021

3-Materials and Methods: where did you perform the clinical examination for the included children? Explain that. Also, explain if any method was followed to ensure randomization of the nurseries included. It seems that you tried to include something about the nurseries but that was in Spanish.

4-Materials and Methods: Measures lines 123-126: description of the questions is mentioned in Spanish.

5-Results: Table 1: it is better that you include all percentages. Also, on what basis did you perform the categorization for socioeconomic level?

6- Results: it is better to justify using 2-way ANOVA in the statistical analysis.

7-Results line 214: recheck the data in this line “(p < 0.05) (x2 (1)=4.43, p=0.03) (p < 0.05) x2 (1)= 6.52, p= 0.01”

8-Line 239 and 266: Student's T test

9-Verification of intra-examiner reliability was not mentioned among the limitations.

Author Response

1-The abstract should be modified according to the changed manuscript (e.g. age group of children and conclusions).

We have modified part of the summary. However, we cannot give more details about the conclusions because there is a word limit that we cannot exceed. Line 15 and Line 25 and 26.

2-Introduction lines 52-54: “However, breastfeeding, apart from the innumerable systemic, psychological, immunological and nutritional benefits [13-18], also stands out for its benefits at the oral level [19].”

Support these sentences with the below recent reference:

Risk Factors of Early Childhood Caries Among Preschool Children in Eastern Saudi Arabia. Science Progress 2021

The reference has been added in line 56.

3-Materials and Methods: where did you perform the clinical examination for the included children? Explain that. Also, explain if any method was followed to ensure randomization of the nurseries included. It seems that you tried to include something about the nurseries but that was in Spanish.

The clinical examination of the children was performed in a dental clinic. Line 136 in the subject and method section has been added.

We have translated how the randomization was performed. Sorry for the inconvenience.

4-Materials and Methods: Measures lines 123-126: description of the questions is mentioned in Spanish.

We have translated it. Sorry for the inconvenience.

5-Results: Table 1: it is better that you include all percentages. Also, on what basis did you perform the categorization for socioeconomic level?

The % symbol has been included. This categorisation recording mixed levels of socio-economic status has been used in other occasions. For example:

Lera Marqués L, Olivares Cortés S, Leyton Dinamarca B, Bustos Zapata N. Dietary patterns and its relation with overweight and obesity in Chilean girls of medium-high socioeconomic level. Arch Latinoam Nutr. 2006 Jun;56(2):165-70. PMID: 17024962.

Álvarez C, Guzmán-Guzmán IP, Latorre-Román PÁ, Párraga-Montilla J, Palomino-Devia C, Reyes-Oyola FA, Paredes-Arévalo L, Leal-Oyarzún M, Obando-Calderón I, Cresp-Barria M, Machuca-Barria C, Peña-Troncoso S, Jerez-Mayorga D, Delgado-Floody P. Association between the Sociodemographic Characteristics of Parents with Health-Related and Lifestyle Markers of Children in Three Different Spanish-Speaking Countries: An Inter-Continental Study at OECD Country Level. Nutrients. 2021 Jul 31;13(8):2672. doi: 10.3390/nu13082672. PMID: 34444832; PMCID: PMC8402068.

6- Results: it is better to justify using 2-way ANOVA in the statistical analysis.

Justification has been added to the statistical analysis section (Line 180-183).

7-Results line 214: recheck the data in this line “(p < 0.05) (x2 (1)=4.43, p=0.03) (p < 0.05) x2 (1)= 6.52, p= 0.01”

There was a spelling error in the text.

8-Line 239 and 266: Student's T test

The translation error has been corrected.

9-Verification of intra-examiner reliability was not mentioned among the limitations.

It has been added as a limitation to the discussion section (Line 369-371).